# ACCURACY OF WHITE BOX AND BLACK BOX ADVERSARIAL ATTACKS ON A SIGN ACTIVATION 01 LOSS NEURAL NETWORK ENSEMBLE

**Yunzhe Xue & Usman Roshan**
Department of Data Science
New Jersey Institute of Technology
Newark, NJ 07102, USA
{yx277,usman}@njit.edu

## ABSTRACT

In this work we ask the question: is an ensemble of single hidden layer sign activation 01 loss networks more robust to white box and black box adversarial attacks than an ensemble of its differentiable counterpart of cross-entropy loss with relu activations and an ensemble of the approximate differentiable counterpart of cross-entropy loss with sign activations? We consider a simple experimental setting of attacking models trained for binary classification on pairwise CIFAR10 datasets - altogether a total of 45 datasets. We study ensembles of **bcebp**: binary cross-entropy loss with relu activations trained with back-propagation, **bceban**: binary cross-entropy loss with sign activations trained with back-propagation with the straight through estimator gradient, **01scd**: 01-loss with sign activations trained with gradient-free stochastic coordinate descent, and **bcescd**: binary cross-entropy loss with relu activation trained with gradient-free stochastic coordinate descent (to isolate the effect of 01 loss from gradient-free training). We train each model in an ensemble with a different random number generator seed. Our four models have similar mean test accuracies in the mid to high 80s on pairwise CIFAR10 datasets but under powerful PGD white-box attacks they each drop to near 0% except for our 01 loss network ensemble that has 31% accuracy. Even training with the gradient-free stochastic coordinate descent can be attacked thus suggesting that the defense lies in 01 loss. In a black-box transfer attack we find adversaries produced from the bcebp model fully transfer to bceban but much less to 01scd - we see the same transferability pattern from bceban to bcebp and 01scd. We also find that adversaries from 01scd barely transfer to bcebp and bceban. While our results are far from those of multi-class and convolutional networks, they suggest that 01 loss models are hard to attack naturally without any adversarial training. All models, data, and code to reproduce results here are available from https://github.com/xyzacademic/mlp01example.

## 1 INTRODUCTION

Adversarial attacks remain a security vulnerability in neural networks today since they were first discovered (Szegedy et al., 2014). A recent paper evaluating the operational feasibility of adversarial attacks in military defense (Zhang et al., 2022) found that patch attacks posed a minimal danger in practice but both white and black-box attacks can be significantly more effective - a white box attack reduces the target model's accuracy by 65% and a black-box attack reduces it by 55% to 63% depending upon knowledge and access to training data and target model architecture. Various defenses have been developed and broken (Athalye et al., 2018). Adversarial training (Kurakin et al., 2016), which is to train the model with clean and adversarial data, remains the most effective solution to date but it is computationally expensive and lowers clean test accuracy (Raghunathan et al., 2019; Clarysse et al., 2022).

Based on intuition and previous work (Xie et al., 2019; Xue et al., 2020b;a; Yang et al., 2020; 2022) we hypothesize that a 01-loss neural network model would be hard to attack. With a new stochastic

coordinate descent algorithm (Xie et al., 2019) we were able to train a single hidden layer 01-loss neural network with accuracy and runtimes comparable to its differentiable and cross-entropy loss counterparts (see Table 1 below).

Table 1: Accuracy of our models on clean test data averaged across all 45 pairs of classes in CIFAR10

|  | Accuracy (mean of all 45 pairs) | | | | Runtime to train one pair (seconds) | | | |
|---|---|---|---|---|---|---|---|---|
| Data | bcebp | bceban | 01scd | bcescd | bcebp | bceban | 01scd | bcescd |
| Train | 94.2 | 94.88 | 88.6 | 92.4 | 40 | 46 | 64 | 56 |
| Test | 87.2 | 85.42 | 83.2 | 86.8 | N/A | N/A | N/A | N/A |

## 2 MAIN RESULT

In a white box attack the adversary has full access to the target model complete with all architecture details and model weights. Both FGSM (Goodfellow et al., 2015) and PGD (Madry et al., 2017) are popular attack methods. In FGSM the adversary uses the target model gradient once ($x' = x + \epsilon sign(\nabla_x \mathcal{L}(\theta, x, y))$) whereas in PGD this is performed iteratively ($x^{t+1} = x^t + \alpha sign(\nabla_x \mathcal{L}(\theta, x^t, y))$). Thus PGD is a more effective attack. We perform 20 steps of PGD with standard parameter settings $\alpha = 2/256$ and $\epsilon = 16/256$.

Since both FGSM and PGD need the target model gradient we approximate the 01 loss gradient with the cross-entropy loss gradient and use the straight through estimator (Courbariaux et al., 2016) for sign activation gradient: $\frac{df}{dx} = 1$ if $|x| \leq \sum_{x' \in X} x'$ and 0 if $|x| > \sum_{x' \in X} x'$. This is the same estimator to train the cross-entropy loss network with sign activation (also known as binary neural network) with back propagation. For each image in each test dataset we produce an adversary with the target model's gradient (averaged across each model in the ensemble) and then evaluate its accuracy on the target model.

In Table 2 we see that both binary cross-entropy models bcebp and bceban trained with back propagation have adversarial accuracy almost 0%. Even bcescd, which is the same model as bcebp but trained with gradient free stochastic coordinate descent, can be attacked (see accuracies of 9.7% and 0.2% in FGSM and PGD attacks below). Thus we conjecture that the 01 loss is the main defense that makes 01scd harder to attack.

Table 2: FGSM and PGD white box accuracy of our models averaged across all 45 pairs of classes in CIFAR10

| FGSM white box accuracy | | | | PGD white box accuracy | | | |
|---|---|---|---|---|---|---|---|
| bcebp | bceban | 01scd | bcescd | bcebp | bceban | 01scd | bcescd |
| 0.8% | 1.4% | 44.2% | 9.7% | 0.2% | 0.1% | 31.4% | 0.2% |

In our black box attacks we produce adversaries targeting a "source" model and evaluate them on target models. This simulates a scenario where the attacker does not have the target model with complete weights but has the same training dataset and same architecture as the target model. In Table 3 we see that adversaries sourced from bcebp and bceban transfer to each other but much less so to 01scd. We also see that adversaries from 01scd barely transfer to bcebp and bceban.

Table 3: PGD black box transfer attack accuracy of our models averaged across all 45 pairs of classes in CIFAR10

| Source | bcebp | | bceban | | 01scd | |
|---|---|---|---|---|---|---|
| Target | bceban | 01scd | bcebp | 01scd | bcebp | bceban |
| | 0.6% | 59.3% | 0.9% | 60.7% | 78.9% | 75% |

URM STATEMENT

The authors acknowledge that at least one key author of this work meets the URM criteria of ICLR 2023 Tiny Papers Track.

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
