# OpenReview forum: "Accuracy of white box and black box adversarial attacks on a sign activation 01 loss neural network ensemble"
_ICLR.cc/2023/TinyPapers — Submitted to Tiny Papers @ ICLR 2023_

### Official Review · Reviewer_UaYt · 2023-03-30

**Confidence:** 5

**Summary Of Contributions:**

This work investigates the robustness of single hidden layer with sign activation 01 loss networks.

**Rating:**

Needs Clarification (NC): a submission which does not meet the reviewing criteria and needs clarification for its described problem or solution

**Strengths And Weaknesses:**

Strengths:

1. The source codes have been provided.

Weaknesses:

1. This is a very unclear paper with poor written, which is hard to follow.
2. The problem explored in this work is not novel. Also, the authors have not shown the necessity of investigating this work.
3. The experimental results are not proficient. The authors need to consider PGD with different perturbation steps or the robustbench benchmark.

**Suggested Changes:**

Please refer to the weaknesses.

---

### Official Review · Reviewer_hkdg · 2023-04-01

**Confidence:** 3

**Summary Of Contributions:**

The authors investigate whether an ensemble 01 loss networks are more robust to white and black-box attacks compared to ensembles of cross-entropy networks. They find that 01 loss models are harder to attack naturally without adversarial training.

**Rating:**

Clear, Correct, and Reproducible (CCR): a submission which meets the reviewing criteria

**Strengths And Weaknesses:**

**Strengths:**

- The authors release their source code for reproduction.
- Interesting, to-the-point approach.

**Weaknesses:**

- Lack of clarity in some sections
- Hard-to-follow paper (See Suggestions)

**Suggested Changes:**

- Solid paper overall, and I'd like to see a longer version with more experiments (with varying parameters) soon.
- I would suggest simplifying the abstract to provide an overview of the paper in a concise manner. A lot of the matter can be shifted to the Introduction section, which seems to be underdeveloped and missing specific details.
- The authors may also like to (explicitly or implicitly) go more into the details of the ensembles themselves for unaware but interested readers. Combined with the dense abstract, it was sometimes hard to follow the authors' proposal.
- (Nitpick) I'd suggest the authors use `\citet{}` and `\citep{}` macros in their work for citations. Or, change the way citations are handled to differentiate the author's name in the citation from the main body of the text.

---

### Comment · Area_Chair_wWpT · 2023-06-05
**Ready to archive**

This work meets the threshold for archival, contains the URM statement, and is deanonymized.

---

> ### Author Response · Authors · 2023-06-15
> **Consent to archive**
>
> I hereby give consent to have my paper archived.

---

### Meta-Review · Area_Chair_wWpT · 2023-04-07

**Recommendation:** Invite to present
**Confidence:** 4

**Metareview:**

Thank you for your submission. The reviewers have noted the conciseness of your approach and found your paper to be overall clear, correct, and reproducible. They reviewers have also noted that to increase the clarity further, some additional clarifications as well as some more details on the evaluation. Overall, these changes appear to be mostly word level so the revision process should be minor.

**Summary:**

This paper explores the adversarial robustness in binary classifications using 01 loss. They find that 01 losses work as somewhat of a natural defense for models that would otherwise be vulnerable to adversarial examples.

**Reason For Not Giving A Higher Recommendation:**

- Needs some additional clarification
- Changes require some simple paper restructuring

**Reason For Not Giving A Lower Recommendation:**

- Clear, correct, and reproducible
- Promising results

---

### Decision · Program_Chairs · 2023-04-08

Invite to present